# Combining Transcriptome- and Metabolome-Analyzed Differentially Expressed Genes and Differential Metabolites in Development Period of Caoyuanheimo-1 (*Agaricus* sp.) from Inner Mongolia, China

Hai-yan Wang [1,2], Ya-nan Lu [2], Ya-jiao Li [2], Guo-qin Sun [2], Yuan Wu [2], Rui-qing Ji [3] and Wei Yan [1,*]

[1] College of Forestry, Inner Mongolia Agricultural University, Hohhot 010031, China; wanghaiyan66@163.com
[2] Vegetable and Flower Research Institute, Inner Mongolia Academy of Agricultural and Animal Husbandry Sciences, Hohhot 010031, China
[3] Engineering Research Center of Edible and Medicinal Fungi, Ministry of Education, Jilin Agricultural University, Changchun 130118, China
* Correspondence: yanwei89911@163.com

**Abstract:** Caoyuanheimo-1 (*Agaricus* sp.) is a delectable mushroom native to Inner Mongolia, China, belonging to the *Agaricus* genus and valued for both its edible and medicinal properties. Although it has been cultivated to a certain extent, the molecular mechanisms regulating its development remain poorly understood. Building on our understanding of its growth and development conditions at various stages, we conducted transcriptomic and metabolomic studies to identify the differentially expressed genes (DEGs) and metabolites throughout its growth cycle. Simultaneously, we analyzed the synthesis pathways and identified several key genes involved in the production of terpenoids, which are secondary metabolites with medicinal value widely found in mushrooms. A total of 6843 unigenes were annotated, and 449 metabolites were detected in our study. Many of these metabolites and differentially expressed genes (DEGs) are involved in the synthesis and metabolism of amino acids, such as arginine, cysteine, methionine, and other amino acids, which indicates that the genes related to amino acid metabolism may play an important role in the fruiting body development of Caoyuanheimo-1. Succinic acid also showed a significant positive correlation with the transcriptional level changes of nine genes, including laccase-1 (TRINITY_DN5510_c0_g1), fruiting body protein SC3 (TRINITY_DN3577_c0_g1), and zinc-binding dihydrogenase (TRINITY_DN2099_c0_g1), etc. Additionally, seventeen terpenoids and terpenoid-related substances were identified, comprising five terpenoid glycosides, three monoterpenoids, two diterpenoids, one sesquiterpenoid, one sesterterpenoid, two terpenoid lactones, and three triterpenoids. The expression levels of the genes related to terpenoid synthesis varied across the three developmental stages.

**Keywords:** primordium period; fruiting body period; differentially expressed genes; metabolites; terpenoids





## 1. Introduction

The Prairie Brown Mushroom, commonly known as Caoyuanheimo, is a delectable type of mushroom characterized by its brown fruiting bodies which belongs to the Agaricus genus. It predominantly grows in grasslands, mainly within the Xilingol League of the Inner Mongolia Autonomous Region [1]. Although it is purported to be found in other parts of Inner Mongolia, including Hulunbuir City, Chifeng City, Hohhot City, and Ordos City, there are no official reports in the literature confirming its distribution in these areas. The taxonomic status of this species has been a subject of controversy. It has been reported to belong to either *Agaricus arvensis* [1,2] or *Agaricus bernardii* [3]. However, morphological observations and preliminary ITS sequence data from our research group suggest that it is more closely related to *Agaricus bisporus*, albeit with some differences. Consequently, its

taxonomic classification requires further confirmation through comprehensive analyses, including additional sequencing, morphological and anatomical characteristics, habitats, and other relevant factors.

This mushroom is highly regarded for its delicious taste and is popular among the local population. However, its wild germplasm resources are diminishing and no longer meet the demands of the people. In response to this situation, our laboratory has conducted extensive research on its cultivation and domestication. Caoyuanheimo-1 (*Agaricus* sp.), derived from the fruiting body of the Prairie Brown Mushroom, is the variety that has been domesticated in our laboratory.

Transcriptomics and metabolomics have demonstrated unique advantages in uncovering the molecular mechanisms underlying the growth and development of edible and medicinal fungi [4–6]. The transition from mycelium growth to reproductive growth is indeed a pivotal phase in the life cycle of edible fungi, often termed the primordium period. During this period, environmental factors such as low temperature, appropriate light exposure, and proper ventilation play crucial roles in initiating the formation of primordia. However, in actuality, the genetic makeup of the fungus plays a substantial role in orchestrating the transition from vegetative growth to reproductive growth, including the development of primordia and subsequent fruiting bodies. Genetic factors regulate the expression of key genes involved in these processes, responding to environmental signals and coordinating the intricate molecular pathways required for reproductive development. Based on a comprehensive analysis of the transcriptome data from 11 fungi along with four related environmental factors, large fungi were concluded to share a common pathway that likely plays a crucial role in primordium formation; for example, the MAPK pathway and the Hedgehog signaling pathway played significant regulatory roles in the light-induced formation of primordia [7,8], while some genes encoding heat shock proteins, the G protein, and β-1,3-glucanase in the GH5 family were involved in the fruiting body development process of *Lentinus edodes* [9].

Studies have shown that the genes involved in the formation of fruiting bodies in *Agaricus bisporus* include the *AtpD* gene encoding an ATP synthase subunit, the *CypA* gene encoding a cytochrome p450, and the *SepA* gene encoding a membrane protein, which are highly expressed during the formation of fruiting bodies [10]. Hydrophobic proteins also play a crucial role in the development of the fruiting body of *Agaricus bisporus* As2796 [11]. Mannitol is the main means of carbon storage in *A. bisporus*, which is of great significance for its growth, fruit development, and stress tolerance. The expression of NADP-dependent mannitol dehydrogenase (*MtDH*), which controls mannitol synthesis under salt stress, was induced, leading to mannitol accumulation [12]. By integrating transcriptomic and metabolomic data, we can gain a holistic understanding of the molecular networks driving the growth and development of Caoyuanheimo-1. This systems biology approach allows for the identification of key genes, enzymes, and metabolites involved in important biological processes, facilitating targeted interventions to improve fungal cultivation, optimize the production of bioactive compounds, and harness the potential of fungi for various biotechnological applications.

Terpenoids are a large family of secondary metabolites widely distributed in nature [13–16]. These compounds and their derivatives, composed of isoprene (C5) units, can be classified into monoterpenes (C10), sesquiterpenes (C15), diterpenes (C20), triterpenes (C30), and polyterpenes based on the number of carbon atoms [17]. Studies have shown that terpenoids in plants have various functions, such as defense, disease resistance, allelopathy, and insect induction [18–22]. Research has also been conducted on terpenoid compounds in fungi [23–27], such as tetracyclines and pleuromutilin (Drosophilin B) [28,29]. In addition, terpenoids have important medicinal functions [30–33]. The biosynthetic pathways of terpenoids have been clearly studied in plants, where terpenoids are synthesized independently through two pathways. One is the membrane-associated pathway (MVA) located within the cytoplasm [34,35], and the other is the 1-deoxy-D-xylose-5-phosphase pathway

(DXP) [36] or methylerythritol 4-phase pathway (MEP) located within the cytoplasm [37]. The biosynthesis of terpenoids in fungi has also been reported on [38–41].

Based on this, genetic factors are assumed to play a key role in the environmental influences on Caoyuanheimo-1 in the process of primordium formation, fruiting body development, and maturation. At the same time, metabolites in different stages of development also show differences, including terpenoids. Therefore, the transcriptomic and metabolomic studies in this article aim to reveal the molecular mechanism of Caoyuanheimo-1's growth and development, which will lay a foundation for its efficient development and utilization.

## 2. Materials and Methods

### 2.1. Sample Preparation

The key technical steps in the cultivation of Caoyuanheimo-1 involve the formation of the primordium and the progression of young fruiting bodies to mature ones.

The culture substrate was a 50 kg culture base formula: dried wheat straw or straw (one or two kinds of mixture), 23 kg; dried cow or goat manure, 22 kg; soybean cake meal or corn meal, 3.5 kg; dipotassium hydrogen phosphate 50 g; diammonium phosphate, 50 g; and white ash, 400 g.

Growth and development process: Initially, the mycelium was cultured in darkness at a controlled temperature range of 20 °C to 23 °C and a relative humidity of approximately 80%. This phase focuses on establishing a healthy mycelial network. After 20–25 days, once the mycelium had sufficiently grown, the culture was covered with soil. The emergence period of the mushrooms requires scattered light, indicating a shift from the previous dark conditions. The temperature remained cool but varied up to 23 °C, and the humidity was slightly higher, at about 85%. This environment supported the initial appearance of mushroom primordia. Primordia (Figure 1a) appeared 17–22 days after soil covering. Shortly thereafter, within 2–3 days, these primordia developed into young fruiting bodies (Figure 1b). The mature fruiting bodies (Figure 1c) were ready for harvest 4–6 days after their formation.

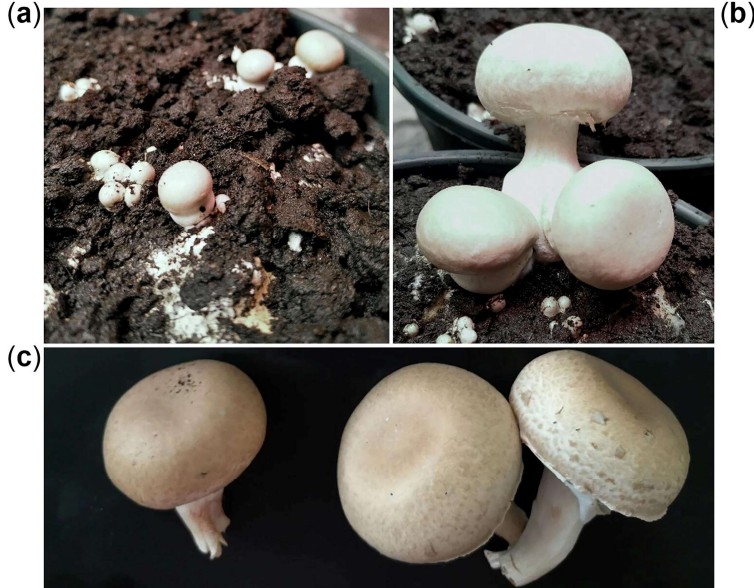

**Figure 1.** The phenotype of Caoyuanheimo-1 (*Agaricus* sp.) and function annotations of transcriptome sequencing. (**a**) A photo of the primordia of Caoyuanheimo-1; (**b**) a photo of the young fruiting bodies of Caoyuanheimo-1; and (**c**) a photo of the ripe fruiting bodies of Caoyuanheimo-1.

Sampling: The samples in this experiment were collected from the full primordium (diameter less than 1 cm: HY), full young fruiting bodies (cap diameter 4.5–5.0 cm: HZ), and full ripe fruiting bodies (cap diameter 7.5–8.5 cm: HC). HY represents the primordium

period of Caoyuanheimo-1 (*Agaricus* sp.) development, HZ represents the young fruiting period of development, and HC represents the ripe fruiting period of development. Three independent biological samples were selected for repeated experiments at each developmental period, with a total of nine samples tested.

### 2.2. LC-MS/MS and PCA Analysis

The untargeted metabolites of Caoyuanheimo-1 (*Agaricus* sp.) at three different developmental periods were explored.

LC-MS/MS analyses were performed using an UHPLC system (Vanquish, Thermo Fisher Scientific, Waltham, MA, USA). The mobile phase used ammonium acetate and acetic acid solution, with an injection volume of 2 μL. The temperature of the automatic sampler was 4 °C.

Xcalibur was used as the acquisition software for collecting the MS/MS spectra and for continuously evaluating the full scan MS spectrum as it could be used to control the QE HFX mass spectrometer.

The principal component analysis (PCA) and OPLC-DA [42] of the experimental samples was performed using the software SIMCA V14.1 (https://landing.umetrics.com/simca-free-trial-offer 15 December 2021) for revealing the internal structure of the data by better explaining the data variables [43]. Compared with PCA, PLS-DA can maximize the distinction between groups, which is conducive to the search for differential metabolites. Orthogonal partial least squares discriminant analysis (OPLS-DA) combines orthogonal signal correction (OSC) and PLS-DA methods, which can decompose the X matrix information into two types of information related to Y and not related to Y, and screen the differential variables by removing the irrelevant differences, which is conducive to searching for differential metabolites.

Based on the results of OPLS-DA, the obtained multivariate analysis of the variable importance in project (VIP) of the OPLS-DA model can initially screen out the metabolites with differences between different varieties or tissues. Then, the $p$-value or fold change of univariate analysis can be combined to further screen out differential metabolites. We adopted the method of combining the $p$-value and VIP value of the OPLS-DA model to screen differential metabolites. The screening criteria for differentially accumulated metabolites were $p$-value (the $p$-value of the Student's $t$-test) < 0.05 and VIP (the variable Importance in the Projection for the first principal component of OPLS-DA model) > 1 (with duplicates), or $p$-value < 0.05 and fold change > 1.2 [44,45].

The preprocessed data were annotated in the HMDB database V4.0 (https://hmdb.ca) [46] and KEGG COMPOUND database (https://www.kegg.jp/kegg/compound/) [45].

### 2.3. Transcriptome Sequencing and Analysis

The transcription profiles of Caoyuanheimo-1 (*Agaricus* sp.) at three different developmental periods were explored.

The purity of the RNA used in the transcriptome sequencing was determined using NanoDrop 2000 (Thermo Fisher Scientific, Wilmington, DE, USA). According to the manufacturer's recommendation, each sample was sequenced using 1 μg of RNA. An NEBNextR UltraTM Direct RNA Library Prep Kit was used with IlluminaR (NEB, USA) for library construction. The purification of the RNA used poly-T oligo-attached magnetic beads; the first-strand cDNA synthesis used M-MuLV Reverse Transcriptase, and the second-strand cDNA synthesis used DNA polymerase I and RNase H. The library fragments were purified using the AMPure XP system (Beckman Coulter, Beverly, MA, USA). The PCR was performed using Phusion High Fidelity DNA polymerase. The PCR product was purified and evaluated for library quality using the Agilent Bioanalyzer 2100 system.

We used Perl scripts to process the raw data; clean data were then obtained. The Q20, Q30, and GC content, and sequence duplication level of the clean data were also calculated. These clean reads were then mapped to the reference genome sequence using the HISAT2 tools software.

The gene expression levels were estimated based on the fragments per kilobase of transcripts per million fragments mapped (FPKM). A differential expression analysis was performed using the DESeq [47,48] R package (1.10.1). We used the site (https://hiplot.com. cn/cloud-tool/drawing-tool/list) to make a Venn diagram showing the number of different genes and shared genes in different groups. The Benjamin and Hochberg's approach was used to adjust the *p*-value and to control the false discovery rate. Genes with an adjusted *p*-value < 0.01 and absolute value of log2(fold change) > 1 found by DESeq were considered differentially expressed.

Gene function was annotated based on the KEGG (Kyoto Encyclopedia of Genes and Genomes) database. The KEGG database is used to search for pathways enriched with differentially expressed genes, as well as which branch genes have differential expressions in each pathway [49]. We used the KOBAS [49] software to test the statistical enrichment of differentially expression genes in the KEGG pathways, and we used clusterProfiler R packages to find the KEGG pathways that were significantly enriched compared with the entire genome background.

### 2.4. Correlation Analysis of Differentially Expressed Genes and Differently Accumulated Metabolites

In order to explore the interaction between differentially expressed genes and metabolites, as well as the differential gene expression related to metabolites at different periods of growth and development, we screened the top 50 genes with the lowest FDR (false discovery rate) values in the transcriptome analysis and the top 50 metabolites with the highest VIP (variable importance in projection) values in the metabolomics analysis for a Spearman correlation analysis. Then, we calculated the Spearman correlation between the two using R (v3.6.2) and visualized it using pheatmap. Afterwards, the significantly (*p* < 0.05) differentially expressed genes were analyzed using a GraphPad prism to draw a bar graph of the expression levels between HY, HZ, and HC.

## 3. Results

### 3.1. Metabolome Analysis of Caoyuanheimo-1 (Agaricus sp.) during growth and Development

A total of 449 metabolites were detected through untargeted metabolomics. These metabolites were classified into 16 super classes, and 18 metabolites were not annotated into categories. Among them, the super class with the highest number of metabolites annotated was lipids and lipid-like molecules, which contained 109 metabolites (Table S1). The results of the principal component analysis (PCA) for differentially accumulated metabolites showed differences in metabolomic data among the three developmental periods (Figure S1).

Eleven differentially accumulated metabolites were found in the HY vs. HZ group, with nine metabolites significantly increasing (*p* < 0.05) in content during the HZ period, including DL-dopa (HMDB0000609), methylglutaric acid (HMDB0061676), D-xylitol (HMDB0002917), uracil (HMDB0000300), pyruvic acid (HMDB0000243), 1,5-anhydrosorbitol (HMDB0002712), S-adenosylhomocysteine (HMDB0000939), oxoadipic acid (HMDB0000225), and 4-[(E)-2-(3,5-dimethoxyphenyl)ethenyl]phenol (HMDB0130987). Additionally, two metabolites significantly decreased (*p* < 0.05) in content during the HZ period: nicotinic acid (HMDB0001488) and urocanic acid (Figure 2a,b). For the HY vs. HZ group, the results showed that the differentially accumulated metabolites were most significantly enriched (*p* < 0.05) for pantothenate and CoA biosynthesis, pentose and glucuronate interconversions, methane metabolism, and the cysteine pathway (Figure 2c).

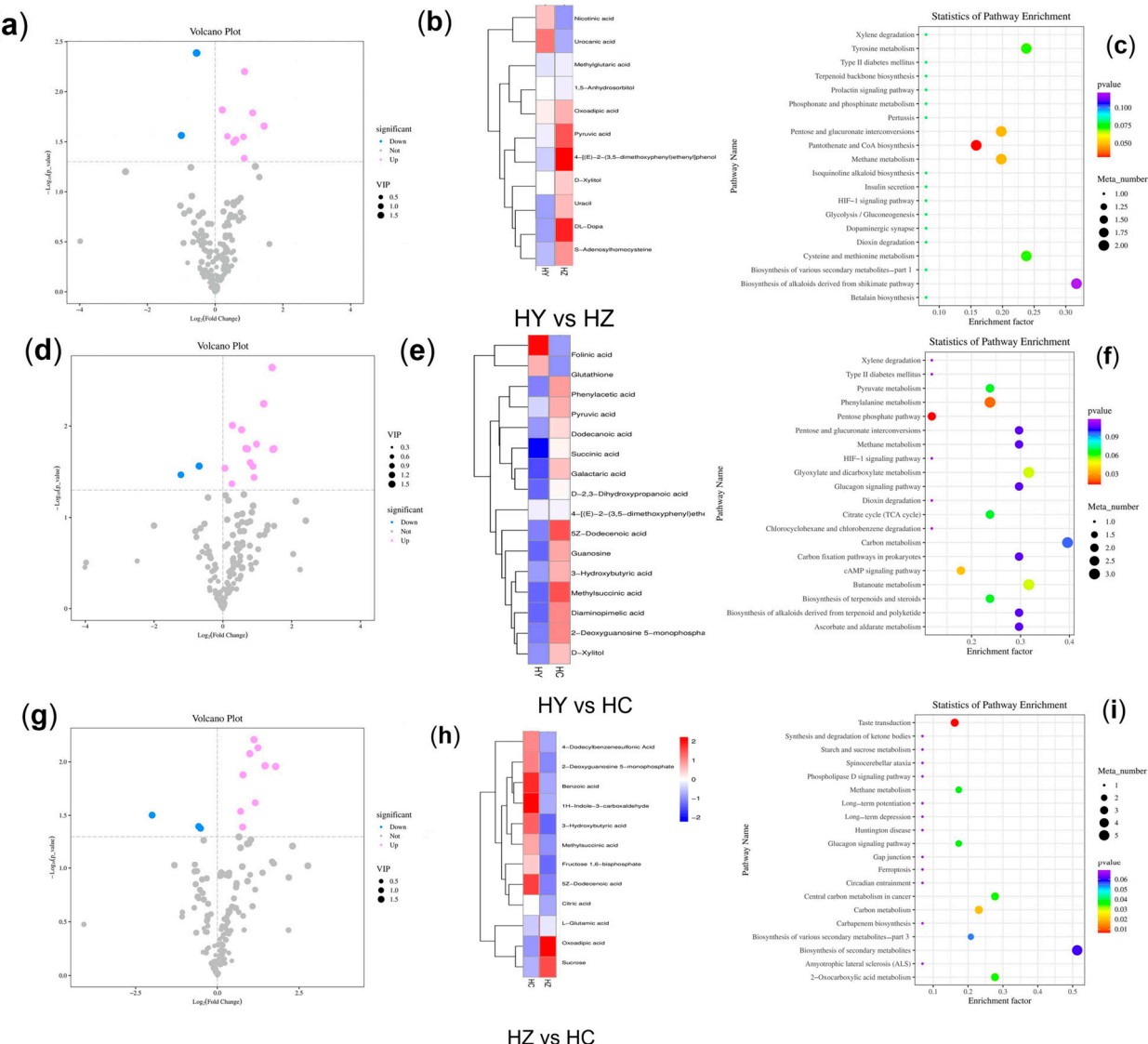

**Figure 2.** A comparative analysis of the different metabolite types in the different developmental stages of Caoyuanheimo−1 (*Agaricus* sp.). HY represents the primordium period of Caoyuanheimo−1, HZ represents the young fruiting period of Caoyuanheimo−1, and HC represents the ripe fruiting period of Caoyuanheimo−1. Volcano plots show the differentially accumulated metabolites between the HY vs. HZ (**a**), HY vs. HC (**d**), and HZ vs. HC (**g**) groups. Each point in the volcano map represents a metabolite; the X axis represents the logarithm of the fold change of a certain metabolite in the two samples; and the Y axis represents the logarithm of the *p*-value. The blue dots represent downregulated differentially accumulated metabolites; the pink dots represent upregulated differentially accumulated metabolites; and gray represents detected but not significantly differentially accumulated metabolites. The size of the dots represents different VIP values. Heatmaps visualization shows the differential metabolites accumulated between the HY vs. HZ (**b**), HY vs. HC (**e**), and HZ vs. HC (**h**) groups. The relative accumulation levels of individual samples are shown; the transition from red to blue represents the level of metabolite accumulation. The differentially accumulated metabolite enrichment analysis of the different development stages are shown as bubble charts of the KEGG enrichment analyses between the HY vs. HZ (**c**), HY vs. HC (**f**), and HZ vs. HC (**i**) groups.

In the HY vs. HC group, there were 16 differential metabolites, of which 14 had a significant increase ($p < 0.05$) in content and 2 had a significant decrease ($p < 0.05$) in content in the HC group. The metabolites with increased levels included 5Z-dodecenoic acid (HMDB0000529),

galactaric acid (HMDB0000639), guanosine (HMDB0000133), diaminopimelic acid (HMDB0001370), 2′-deoxyguanosine 5′-monophosphate (HMDB0001044), 3-hydroxybutyric acid (HMDB0000357), dodecenoic acid (HMDB0000638), D-xylitol (HMDB0002917), methyl-succinic acid (HMDB0001844), phenylacetic acid (HMDB0000209), pyruvic acid (HMDB0000243), 4-[(E)-2-(3,5-dimethoxyphenyl)ethenyl]phenol (HMDB0130987), succinic acid (HMDB0000254), and D-2,3-dihydroxypropanoic acid (HMDB0031818). The metabolites with decreased levels included folinic acid (HMDB0001562) and glutathione (HMDB0000125) (Figure 2d,e). For the HY vs. HC group, the results showed that the differentially accumulated metabolites were most significantly enriched for the pentose phosphate pathway, phenylalanine metabolism, and the cAMP signaling pathway (Figure 2f).

In the HZ vs. HC group, there were 12 differential metabolites, of which 9 had a significant increase ($p < 0.05$) in content and 3 had a significant decrease ($p < 0.05$) in content in the HC group. The metabolites with increased levels included 5Z-dodecenoic acid (HMDB0000529), fructose 1,6-bisphosphate (HMDB0001058), 2′-deoxyguanosine 5′-monophosphate (HMDB0001044), 3-hydroxybutyric acid (HMDB0000357), citric acid (HMDB0000094), methylsuccinic acid (HMDB0001844), 4-dodecylbenzenesulfonic acid (HMDB0059915), benzoic acid (HMDB0001870), and 1H-indole-3-carboxaldehyde (HMDB0029737). The metabolites with decreased levels included L-glutamic acid (HMDB0000148), oxoadipic acid (HMDB0000225), and sucrose (HMDB0000258) (Figure 2g,h). For the HZ vs. HC group, the results showed that the differentially accumulated metabolites were most significantly enriched for taste transduction, carbon metabolism, methane metabolism, the glucagon signaling pathway, and 2-oxocarboxylic acid metabolism (Figure 2i).

The differently accumulated metabolites commonly found in the HY vs. HZ and HY vs. HC groups included three metabolites—D-xylitol (HMDB0002917), pyruvic acid (HMDB0000243), and 4-[(E)-2-(3,5-dimethylphenyl)ether]phenol (HMDB0130987)—the content of which significantly increased ($p < 0.05$) in both the HZ and HC groups. The accumulated metabolites' content, including that of 5Z-dodecenoic acid (HMDB0000529), 2′-deoxyguanosine 5′-monophosphate (HMDB0001044), and 3-hydroxybutyric acid (HMDB0000357), significantly increased ($p < 0.05$) in the HC group. The differently accumulated metabolites commonly found in the HY vs. HZ and HZ vs. HC groups included only one metabolite, oxoadipic acid, whose content significantly increased ($p < 0.05$) in the HZ group but decreased in the HC group. Nicotinic acid (HMDB0001488), urocanic acid, folinic acid (HMDB0001562), and glutathione (HMDB0000125) were found to be differential metabolites of the primordium.

*3.2. Transcriptome Analysis of Caoyuanheimo-1 (Agaricus sp.) during Growth and Development*

A total of 50,303 transcripts and 8208 unigenes were obtained through transcriptome assembly (Table S2). The N50 of the transcripts and unigenes were 4761 bp and 4404 bp, respectively. Among them, the number of transcripts and unigenes with a length greater than 2000 bp were 28,320 and 3586, accounting for 56.30% and 43.69%, respectively (Figure S2). A total of 66.78 Gb of clean data was obtained, with an average of 7.42 Gb of clean data per sample, and the average percentage of Q30 was 92.39% (Table S3). A total of 6843 unigenes were annotated in all databases, including COG (2533), GO (3831), KEGG (4768), KOG (3293), Pfam (3967), Swissprot (4053), and NR (6807). In all databases, the proportion of annotated genes with a length greater than 1000 bp was over 74.23% (Figure 3a). For the species distribution of the top BLAST hits in the NR database, 6564 (96.43%) annotated unigenes matched the sequence of *Agaricus bisporus* (Figure 3b). Differentially expressed genes (DEGs) from three different developmental periods were analyzed for the principal component analysis (PCA); the results showed that the primordium period (HY) was separated from the fruiting period (HZ and HC), viewed on the horizontal axis, and the two periods for the fruiting body converged (Figure S3).

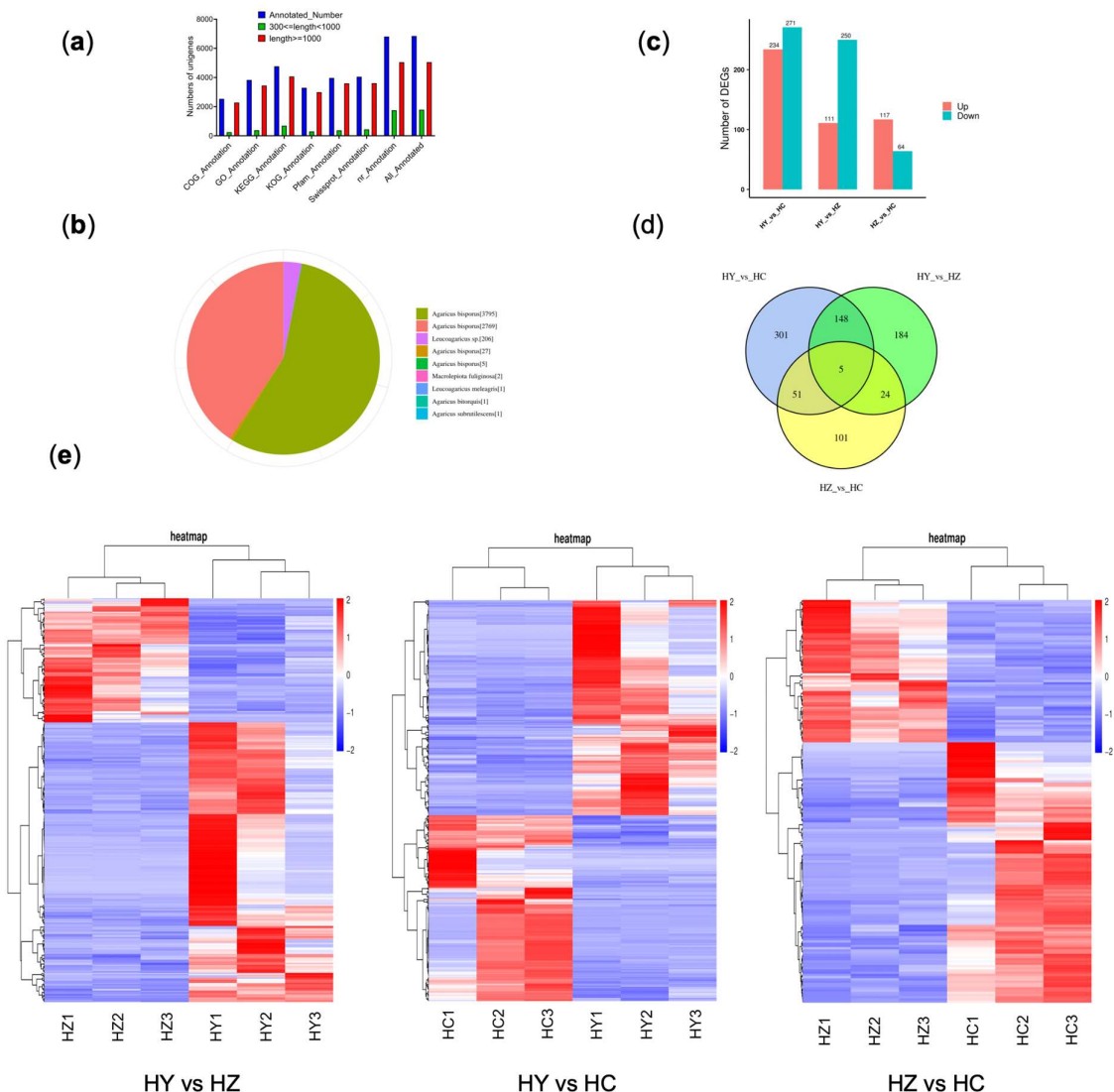

**Figure 3.** A differential gene analysis between the different development stages: HY represents the pri-
mordium period of Caoyuanheimo−1; HZ represents the young fruiting period of Caoyuanheimo−1;
and HC represents the ripe fruiting period of Caoyuanheimo−1. (**a**) Functional annotation numbers
of the unigenes in the COG, GO, KEGG, KOG, Pfam, Swissprot, and NR databases. (**b**) An annotated
species distribution in the NR database. (**c**) A statistical analysis of the number of upregulated and
downregulated DEGs in the HY vs. HC, HY vs. HZ, and HZ vs. HC groups, respectively. (**d**) A Venn
diagram showing the shared and unique genes among the HY vs. HZ, HY vs. HC, and HZ vs. HC
groups. The numbers represent the number of genes that were unique or shared. (**e**) The heatmap
is linked by a dendrogram representing clustering of the transcriptomic data. The color code is as
follows: red indicates upregulated transcripts; blue indicates downregulated transcripts; and white
indicates unchanged transcripts. The color scale of the heatmap ranges from blue (−2.0) to red (2.0)
in the natural logarithmic scale.

By analyzing the HY vs. HC group data, the results showed a total of 505 DEGs, of
which 234 were upregulated and 271 were downregulated in the ripe fruiting body period
(HC) (Figure 3c). Further analysis revealed that the pathways mainly enriched by the
234 upregulated genes included protein processing in endoplasmic reticulum; pheny-
lalanine, tyrosine, and tryptophan biosynthesis; ribosome and N-glycan biosynthe-
sis; MAPK signaling pathway—yeast; beta-alanine metabolism; various types of N-
glycan biosynthesis; tyrosine metabolism; terpenoid backbone biosynthesis; and tryp-
tophan metabolism. The main pathways enriched by the 271 downregulated genes in-

cluded ABC transporters; the biosynthesis of unsaturated fatty acids, glycosphingolipid biosynthesis—globo and isoglobo series; D-arginine and D-ornithine metabolism; caffeine metabolism; glycerolipid metabolism; ascorbate and aldarate metabolism; sphingolipid metabolism; peroxisome; glutathione metabolism; galactose metabolism; alpha-linolenic acid metabolism; fatty acid degradation; arginine and proline metabolism; endocytosis; mitophagy—yeast; methane metabolism; valine, leucine, and isoleucine degradation; and ubiquinone and other terpenoid-quinone biosynthesis (Figure S4).

In the HZ vs. HC groups, there was a total of 181 DEGs, of which 117 were upregulated and 64 were downregulated (Figure 3c). Further analysis revealed that the pathways mainly enriched by the 117 upregulated genes included the pentose phosphate pathway; phosphonate and phosphinate metabolism; selenocompound metabolism; folate biosynthesis; nicotinate and nicotinamide metabolism; tryptophan metabolism; glycerophospholipid metabolism; glyoxylate and dicarboxylate metabolism; glycine, serine, and threonine metabolism; MAPK signaling pathway—yeast; glycolysis/gluconeogenesis; and pyruvate metabolism. The main pathways enriched by the 64 downregulated genes included protein processing in the endoplasmic reticulum and tyrosine metabolism (Figure S5).

In the HY vs. HZ group, there were a total of 361 DEGs, of which 111 were upregulated and 250 were downregulated (Figure 3c). Further analysis revealed that the 111 upregulated genes mainly involved two pathways: protein processing in the endoplasmic reticulum and N-glycan biosynthesis. The main pathways enriched by the 250 downregulated genes included valine, leucine, and isoleucine degradation; ubiquinone and other terpenoid-quinone biosynthesis; tryptophan metabolism; sphingolipid metabolism and selenocompound metabolism; peroxisome Nicotinate and nicotinamide metabolism; nitrogen metabolism; N-glycan biosynthesis; mitophagy—yeast; MAPK signaling pathway—yeast; glyoxylate and dicarboxylate metabolism; glycosphingolipid biosynthesis—globo and isoglobo series; glycolysis/gluconeogenesis; glycerolipid metabolism; glutathione metabolism; galactose metabolism; cysteine and methionine metabolism; caffeine metabolism; and biosynthesis of unsaturated fatty acids (Figure S6).

We analyzed the differentially expressed genes among each group, and the results showed that 153 identical genes differentially expressed in the HY vs. HC and HY vs. HZ groups, 56 identical genes differentially expressed in the HY vs. HC and HZ vs. HC groups, and 29 identical differentially expressed genes in the HY vs. HZ and HZ vs. HC groups (Figure 3c,d). Five differential genes (TRINITY_DN1021_c0_g1, TRINITY_DN1886_c0_g1, TRINITY_DN596_c0_g2, and TRINITY_DN5042_c0_g1, and TRINITY_DN4766_c0_g1) were shared among the three groups. The gene TRINI-TY_DN1021_c0_g1 was annotated from the KOG database, which is a cellular component for primary function, and the endoplasmic reticulum (GO:0005783) and anchored component of the external side of the plasma membrane (GO:0031362) are the cellular components for secondary function.

*3.3. Correlation Analysis between DEGs and Differently Accumulated Metabolites*

The association analysis between differently accumulated metabolites and DEGs revealed some genes closely related to changes in the metabolite content (Figure 4a).

The content of the metabolite polyphobilinogen had a significant positive correlation with transcriptional level changes in 19 genes [including the carboxypeptidase S gene (TRINITY-DN96_c0_g1), allergen Asp f 7 homolog gene (TRINITY-DN4139_c0_g1), probable amino acid permutation PB1C11.02 (TRINITY-DN4571_c0_g1), *O*-methyltransfer tpcA (TRINITY-DN3843_c0_g1), methylthiobulose-1-phosphate dehydrogenase (TRINITY-DN3976_c0_g1), zinc ribbon (TRINITY-DN564_c0_g1), NADH ubiquinone oxidoreductase chain 6 (TRINITY-DN1617_c0_g1), TPX2 (TRINITY-DN1024_c0_g1), and metacaspase-1 (TRINITY-DN5357_c0_g1), etc.] and a significant negative correlation with transcriptional level changes in 5 genes, including mono- and diacylglycerol lipase (TRINITY-DN3483_c0_g1), RNA-dependent RNA polymerase (TRINITY-DN479_c0_g1), etc. (Figure 4a).

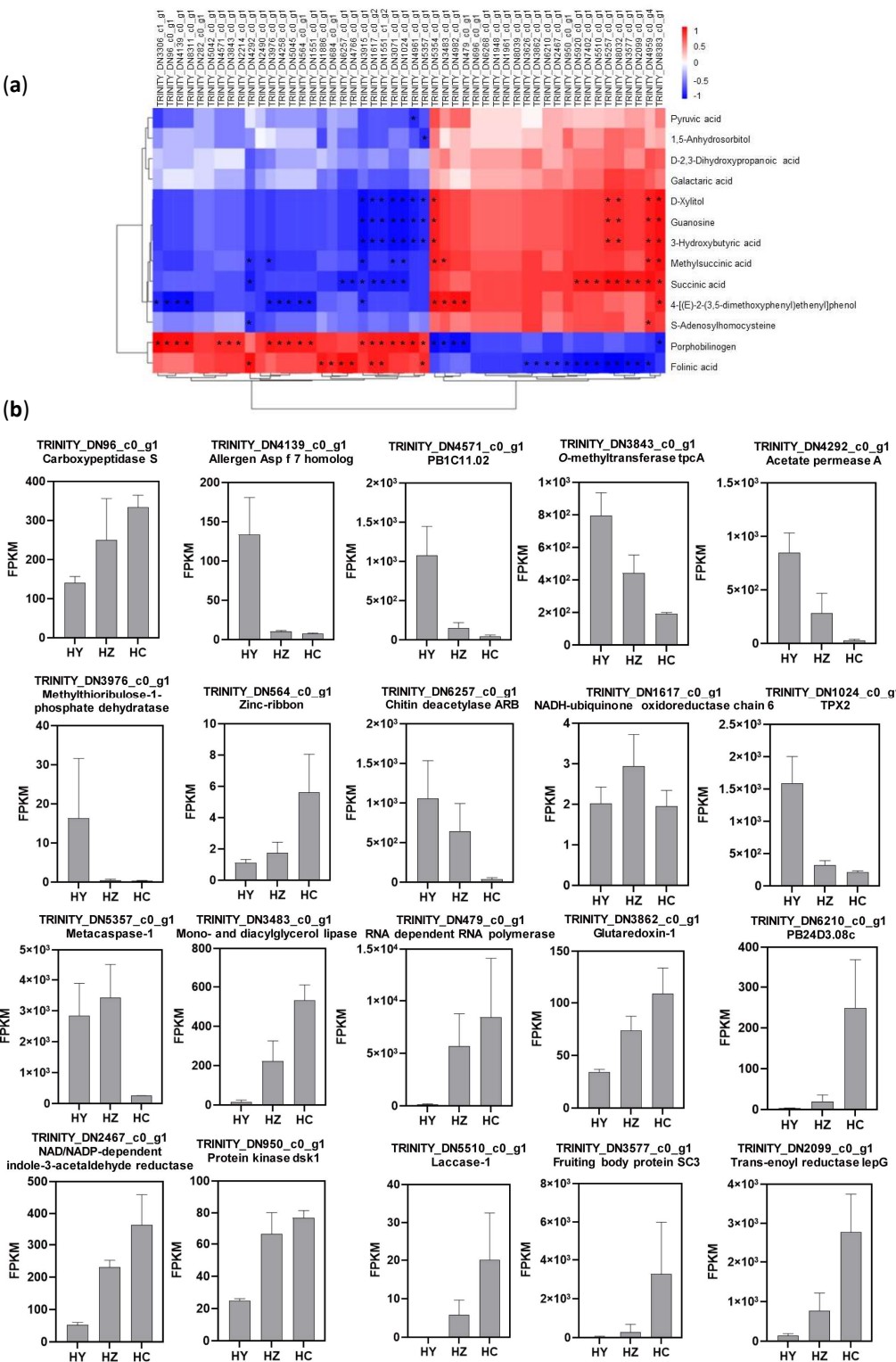

**Figure 4.** The association between differentially expressed genes' data with differentially accumulated metabolites' data. (**a**) A heatmap showing the association of the differentially expressed genes' data with the differentially accumulated metabolites' data. (* indicates statistical significance at $p < 0.05$). The scale of the heatmap represents the Spearman correlation, ranging from blue (−1) to red (1). (**b**) The expression levels of the annotated functional genes at different developmental stages; HY represents the primordium period of Caoyuanheimo−1; HZ represents the young fruiting period of Caoyuanheimo−1; and HC represents the ripe fruiting period of Caoyuanheimo−1.

Succinic acid had a significant positive correlation with the transcriptional level changes of nine genes, including laccase-1 (TRINITY_DN5510_c0_g1), fruiting body protein SC3 (TRINITY_DN3577_c0_g1), and zinc-binding dihydrogenase (TRINITY_DN2099_c0_g1), and a significant negative correlation with the transcriptional level changes of eight genes, including acetate permase A (TRINITY_DN4292_c0_g1), chitin deacetylase ARB (TRINITY_DN6257_c0_g1), NADH ubiquinone oxidoreductase chain 6 (TRINITY_DN1617_c0_g1), and TPX2 (TRINITY_DN1024_c0_g1) (Figure 4a).

Metabolite 4-[(E)-2-(3,5-dimethylphenyl)ether]phenol had a significant positive correlation with transcriptional level changes in 5 genes, including mono- and diacylglycol lipase (TRINITY-DN3483_c0_g1), etc., and had a significant negative correlation with transcriptional level changes in 10 genes, including the carboxypeptidase S gene (TRINITY-DN96_c0_g1), methylthiobulose-1-phosphate dehydrogenase (TRINITY-DN3976_c0_g1), and zinc ribbon (TRINITY-DN564_c0_g1), etc. (Figure 4a).

The metabolite methylsuccinic acid had a significant positive correlation with changes in the transcription levels of four genes, including mono- and diacylglycerol lipase (TRINITY_DN3483_c0_g1). It had a significant negative correlation with changes in the transcription levels of five genes, including acetate permease A (TRINITY_DN4292_c0_g1), methylthiobulose-1-phosphate dehydrogenase (TRINITY_DN3976_c0_g1), and TPX2 (TRINITY_DN1024_c0_g1).

There was a significant positive correlation between the content of metabolite Folinic acid and the transcriptional level changes of 8 genes, including acetate permease A(TRINITY_DN4292_c0_g1), chitin deacetylase ARB (TRINITY_DN6257_c0_g1), NADH-ubiquinone oxidoreductase chain 6 (TRINITY_DN1617_c0_g1), and metacaspase-1 (TRINITY_DN5357_c0_g1), etc., and there was a significant negative correlation with the transcriptional level changes of 13 genes, including glutaredoxin-1 (TRINITY_DN3862_c0_g1), zinc-type alcohol dehydrogenase-like protein PB24D3.08c (TRINITY_DN6210_c0_g1), NAD/NADP-dependent indole-3-acetaldehyde reductase (TRINITY_DN2467_c0_g1), protein kinase dsk1 (TRINITY_DN950_c0_g1), laccase-1 (TRINITY_DN5510_c0_g1), fruiting body protein SC3 (TRINITY_DN3577_c0_g1), and trans-enoyl reductase lepG (TRINITY_DN2099_c0_g1), etc. (Figure 4a).

Three metabolites—D-xylitol, guanosine, and 3-hydroxybutyric acid—all showed significant positive correlations with changes in the transcriptional levels of five genes (no annotation information was available). Additionally, they had significant negative correlations with changes in the transcriptional levels of seven genes, including NADH-ubiquinone oxidoreductase chain (TRINITY_DN1617_c0_g1), TPX2 (TRINITY_DN1024_c0_g1), and metacaspase-1 (TRINITY_DN5357_c0_g1) (Figure 4a); the other four genes had no annotation information.

The carboxypeptidase S gene (TRINITY_DN96_c0_g1), belonging to the Peptidase family M20/M25/M40, consistently upregulated the transcription levels from the primordium to ripe fruiting periods. In addition to this, 10 other genes, including TRINITY_DNS64_C0_g1, TRNITY_DN3483_0_g1, TRINTY_DN79_c0_g1, TRINITY_DN3862_c0_g1, TRINITY_DNS210_c0_g1, TRINITY_DN2467_0_g1, TRINITY_DN950_0_g1, TRINITY_DNS510_C0_g1, TRINITY DN3577_c0_g1, and TRINITY DN2099_c0_g1, all showed consistently upregulated transcription levels from the primordium to ripe fruiting periods (Figure 4b). Seven genes showed consistently downregulated transcription levels from the primordium to ripe fruiting periods, including TRINITY_DN4139_c0_g1, TRINITY_DN4571_C0_g1, TRNITY_DN3843_c0_g1, TRINITY_DN292_c0_g1, TRINTY_DN3976_c0_g1, TRINTY_DN3976_c0_g1, TRINITY_ON1024_c0_g1, and TRINITY_DNS257_c0_g1 (Figure 4b).

### 3.4. Terpenoid Biosynthesis Pathway Analysis

In the development of Caoyuanheimo-1, with some key periods including the primordium, young fruiting body, and ripe fruiting body periods, we detected some genes and pathways of the terpenoids or terpenoid compounds. During the analysis of the metabolomic data, a total of 17 terpenoids were identified, including 5 terpenoid glycosides

(tsangane L 3-glucoside, foeniculoside VII, kenposide B, perilloside C, and (1S,2R,4R)-p-menth-8-ene-2,10-diol 2-glucoside); 3 monoterpenoids (linalyl benzoate, (R)-2-acetoxy-p-mentha-1,8-diene, and crispane); 2 diterpenoids (6,10,14-trimethyl-5,9,13-pentadecatrien-2-one and sclareol); 1 sesquiterpenoid (8, 12-epoxy-4(15), 7, 11-eudesmatrien-1-one); 1 sesterterpenoid (25-cinnamoyl-vulgaroside); 2 terpenoid lactones (eremopetasitenin C2 and lactupicrin); and 3 triterpenoids (ganoderiol H, lucidenic acid A, and lucidenic acid N). The content of these substances gradually accumulates during growth and development (Figure 5a).

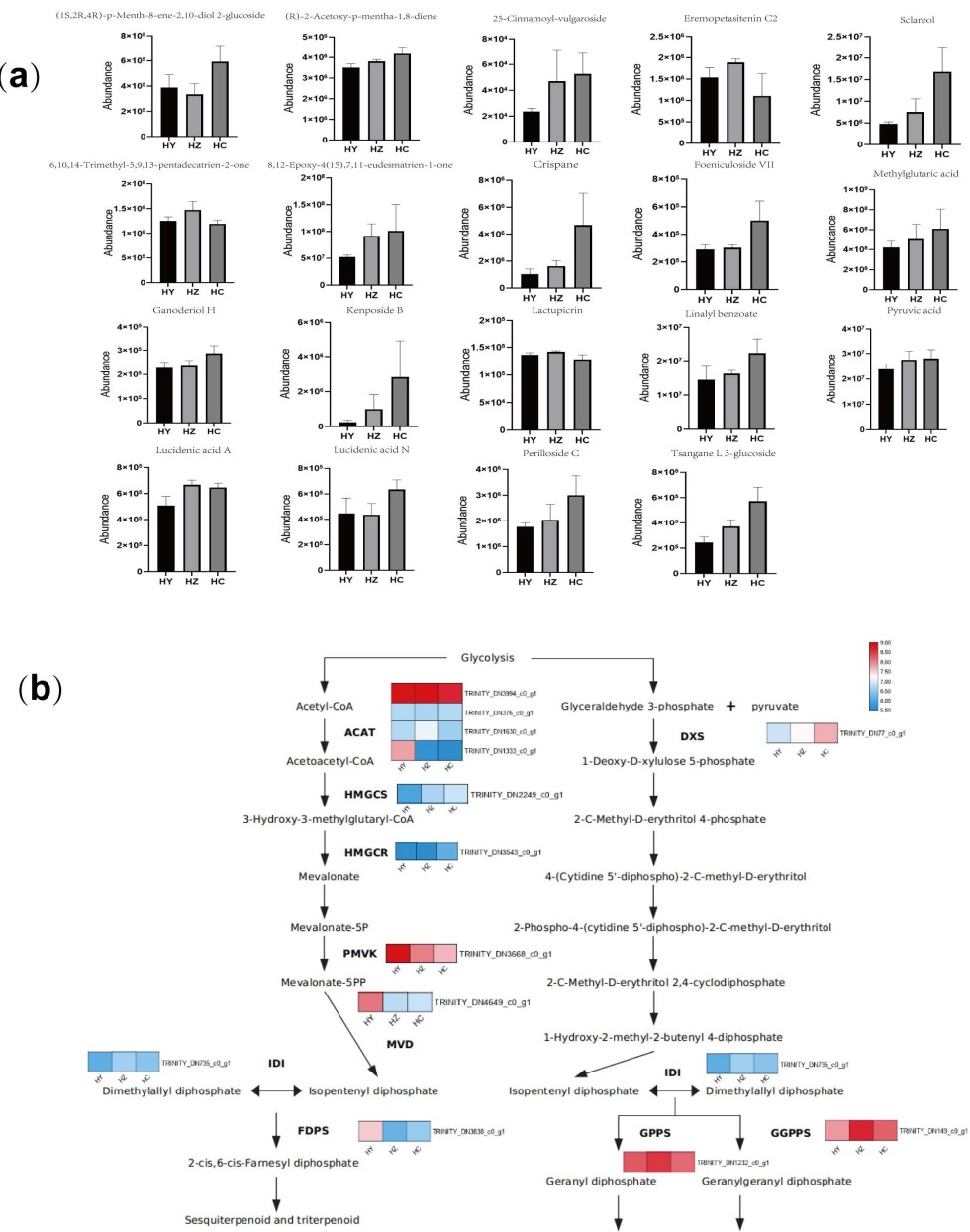

**Figure 5.** An analysis of the key genes in the synthesis pathway of terpenoids and the content of terpenoids during growth and development. (**a**) The bar chart represents the content of terpenoid metabolites and substrates at three different developmental stages; HY represents the primordium period of Caoyuanheimo-1; HZ represents the young fruiting period of Caoyuanheimo-1; and HC represents the ripe fruiting period of Caoyuanheimo-1. (**b**) The heatmap represents the expression levels of key genes in the terpenoid biosynthesis pathway in the HY, HZ, and HC groups. The scale of the heatmap represents the FPKM of genes.

In addition, we found that two differential metabolites were associated with the biosynthesis of terpenoids: pyruvic acid and methylglutaric acid. Moreover, the content of both was at a relatively low level during the HY period and began to increase significantly during the HZ period, reaching its highest value during the HC period (Figure 5a). The results indicate that the precursor content of some terpenoid compounds increases sharply in the HZ and HC periods of development. We also analyzed the transcription levels of various genes in the terpenoid synthesis pathway at different periods and found that the transcription levels of some key genes also underwent significant changes. For example, the transcription level of the DXS (1-deoxy-D-xylulose-5-phase synthesis) gene significantly increased during the HZ and HC periods. The transcription levels of the GGPPS (geranylgeranyl diphosphate synthesis) gene and GPPS (geranyl diphosphate synthesis) gene significantly increased during the HZ phase and decreased to the original level during the HC period (Figure 5b).

## 4. Discussion

Caoyuanheimo is a unique edible mushroom in the Inner Mongolian grasslands. It has much similarity to *Agaricus bisporus*, but it also differs from *Agaricus bisporus*. A previous study named it *Agaricus bernardii* or *Agaricus arvensis* [1–3]. However, its taxonomic status still needs to be confirmed with more in-depth morphological and multi-gene tree construction methods. Our results showed that the transcriptome data from 6564 (96.43%) annotated unigenes matched the sequence of *Agaricus bisporus*, which will provide a reference for resolving the species attribution.

We performed transcriptomic and metabolomic analyses of the three developmental periods (primordium period, young fruiting period, and ripe fruiting period) of Caoyuanheimo-1. Nicotinic acid (HMDB0001488), urocanic acid, folinic acid (HMDB0001562), and glutathione (HMDB0000125) were found to be the differential metabolites in the primordium period, which are B vitamins compounds that could promote cell regeneration and can promote mycelium growth and the primordium development of edible fungi [50]; thus, they may be key metabolic pathways for primordium formation, but further experimental verification is required. Glutathione (HMDB0000125) is a major antioxidant in all forms of life and an indicator of cellular oxidative stress [51]. We analyzed the differentially expressed genes among each group, and the results showed that there were 148 identical differentially expressed genes in the primordium period, which enriched different pathways. Among them, ATP-binding cassette (ABC) transporters were highly enriched in the primordium period. ABC transporters constitute a ubiquitous superfamily of integral membrane proteins that are responsible for the ATP-powered translocation of many substrates across membranes. The highly conserved ABC domains of ABC transporters provide the nucleotide-dependent engine that drives transport [52].

Compared with the primordium period (HY) of Caoyuanheimo-1, the metabolites of methylglutaric acid (HMDB0061676), D-xylitol (HMDB0002917), uracil (HMDB0000300), pyruvic acid (HMDB0000243), and 4-[(E)-2-(3,5-dimethoxyphenyl)ethenyl]phenol (HMDB0130987) showed significant increases ($p < 0.05$) in the fruiting body periods (for both the HZ and HC groups). Additionally, the protein processing pathway in the endoplasmic reticulum and the pentose phosphate pathway were more significant metabolic pathways. Between the two fruiting body periods, the young fruiting period and the ripe fruiting period, the levels of the metabolites L-glutamic acid (HMDB0000148), oxoadipic acid (HMDB0000225), and sucrose (HMDB0000258) decreased in the HZ group but increased in the ripe fruiting period. One hundred eighty-one identical genes were differentially expressed in the young fruiting body period and ripe fruiting body period, which involve the synthesis and metabolism of various amino acids, such as arginine, cysteine, methionine, and other amino acids. These results indicate that the fruiting body development of Caoyuanheimo-1 requires a series of metabolic reactions and that the genes related to amino acid metabolism may play an important role in the fruiting body development, which is consistent with the results of *A. bispora* [53]. The ripening process of

the fruiting body includes spore production and growth, and the increase in carbon and energy metabolism represents vigorous growth and development [54,55]. Furthermore, the processes of cellular respiration and $CO_2$ accumulation play significant roles in influencing the acid-base balance within cells, thereby impacting cellular functions. In the cultivation of *Agaricus bisporus*, the escalating $CO_2$ concentrations during mycelial growth contribute to the gradual acidification of the growth medium [56]. In our experiment, it was also found that the substrate acidity would gradually increase during the growth process, which was similar to that of *Agaricus bisporus*. Additionally, heightened $CO_2$ levels can activate aerobic respiration to eliminate surplus $CO_2$, potentially enhancing ATP production and the generation of respiratory metabolites.

When analyzing the relationship between differential metabolites and differentially expressed genes, we found that for the same metabolite, multiple genes had significant positive and negative correlations; these genes with positive and negative regulatory relationships may have some regulatory relationship between them, which would require further research to confirm. Furthermore, the same gene can also regulate different metabolites. The genes TRINITY_DN4959_c0_g4 and TRINITY_DN8383_c0_g1 had significant correlations with D-xylitol, guanosine, and 3-hydroxybutyric acid, 4-[(E)-2-(3,5-dimethylphenyl)ether]phenol, methylsuccinic acid, and succinic acid but had negative correlations with folinic acid and polyphobilinogen. Interestingly, the metabolites D-xylitol, guanosine, and 3-hydroxybutyric acid, had the same genes (none of these genes had annotated information) associated with them. 3-hydroxybutyrate (3-HB), as a very important metabolite, is found in animals, bacteria, and plants. It is well known that in animals, 3-HB is formed as a product of the normal metabolism of fatty acid oxidation and can therefore be used as an energy source in the absence of sufficient blood glucose. In microorganisms, 3-HB mainly serves as a substrate for the synthesis of polyhydroxybutyrate (PHB), which is a reserve material [57]. However, some fungi have shown the ability to cause PHB degradation [58]. Recent studies have shown that in plants, 3-HB acts as a regulatory molecule that most likely influences the expression of genes involved in DNA methylation, thereby altering DNA methylation levels. Additionally, in animals, 3-HB is not only an intermediate metabolite but also an important regulatory molecule that can influence gene expression, lipid metabolism, neuronal function, and overall metabolic rate [59].

The significant increase in the content of pyruvic acid and methylglutaric suggests that the synthesis of terpenoids in Caoyuanheimo-1 (*Agaricus* sp.) may be related to the accumulation of these two substances. Terpenoids in fungi are generally synthesized in the mevalonate pathway (MVA) [60,61]. Some studies have indicated that the composition of triterpenes contents changes during the fruiting of *Ganoderma lucidum* and that triterpene is synthesized based on acetyl CoA from central carbohydrate catabolism [62]. The MVA is the synthesis of isoprene pyrophosphate and dimethylallyl pyrophosphate from acetyl CoA in eukaryotes. These two conclusions do not contradict each other. The content of 11 terpenoids also consistently increased as the fruiting body matured. Therefore, triterpenoids may be associated with the fructification of edible mushrooms [63].

Two types of terpenoids had lower contents in the young period compared with the primordium period. These two types of terpenoids may be metabolized during the young fruiting period and may be involved in the growth and development process. Three terpenoids also had the lowest contents during the ripe fruiting period, which was also observed in *Ganoderma lucidum* [62], indicating that all three terpenoids may be synthesized or metabolized throughout the entire growth and development process; this lays the foundation for the excavation of terpenoids in fungal mycelia.

**Supplementary Materials:** The following supporting information can be downloaded at: https://www.mdpi.com/article/10.3390/horticulturae10050469/s1; Figure S1. Metabolome PCA analysis of HY and HZ (a), HY and HC (b), and HZ and HC (c). Figure S2. Statistics for the transcripts and unigenes in the transcriptome data. Figure S3. Transcriptome PCA analysis of HY, HZ, and HC. Figure S4. Differentially expressed gene (DEG) KEGG enrichment analysis of HY vs. HC. Figure S5. Differentially expressed gene (DEG) KEGG enrichment analysis of HZ vs. HC. Figure S6. Differentially expressed gene (DEG) KEGG

enrichment analysis of HY vs. HZ. Table S1. Metabolites annotation. Table S2. Length distribution of unigene and transcript .perf. Table S3. Transcriptome sequencing quality control analysis.

**Author Contributions:** W.Y. designed the research plan; H.-y.W. was responsible for the data detection, data analysis, and experimental analysis, etc.; Y.-n.L. was responsible for writing the manuscript; G.-q.S. was responsible for the data analysis; Y.-j.L. was responsible for sample preparation; Y.W. was responsible for the experimental analysis; and R.-q.J. was responsible for the data analysis and manuscript revision. All authors have read and agreed to the published version of the manuscript.

**Funding:** This work was supported by the earmarked fund for CARS20 and Inner Mongolia Agriculture and Animal Husbandry Innovation Fund Project (2022CXJJN01-1).

**Data Availability Statement:** Data are contained within the article and Supplementary Materials.

**Acknowledgments:** Thank you to the guidance teacher for their support and to the partners who have contributed to this work.

**Conflicts of Interest:** The authors declare that they have no competing interests.

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
