# Peer review of "Combining Transcriptome- and Metabolome-Analyzed Differentially Expressed Genes and Differential Metabolites in Development Period of Caoyuanheimo-1 (Agaricus sp.) from Inner Mongolia, China"

_horticulturae, doi:10.3390/horticulturae10050469_

Round 1
Reviewer 1 Report
Comments and Suggestions for Authors
The manuscript sounds to be interesting but English quality is too low. Hence, it is very difficult to evaluate. The objective is not explicit, statistical analyses appears to be wrong (for instance, Principal Components Analysis is claimed to have an associated p-value while it is not, correlation between transcriptome and metabolic data was assessed: just Spearman coefficients without previously verifying biological functions of differentially expressed genes?, in Result some Venn diagrams are shown without being described in Material and Methods how they were constructed, are mentioned of examples of important failures this manuscript presents). Nevertheless, I think many of these failures are due to the bad quality of English language and the scientific approach for writing the manuscript. I consider not also native English but also a specialist in Science communication ought to revise a new version of the research. The latter professional must guide authors in organizing the manuscript, particularly introduction with objectives, material and methods, discussion and conclusion sections. Also, the are some space amnong words that lack and others left over.
Comments on the Quality of English Language
It is too poor.
Author Response
Review #1
Question 1
The manuscript sounds to be interesting but English quality is too low. Hence, it is very difficult to evaluate.
Response: Sorry, we have made corresponding changes in the readability and English writing of the article.
Question 2
The objective is not explicit, statistical analyses appears to be wrong (for instance, Principal Components Analysis is claimed to have an associated p-value while it is not, correlation between transcriptome and metabolic data was assessed: just Spearman coefficients without previously verifying biological functions of differentially expressed genes?
Response: We have reformulated the objective, and corrected the mistakes. And we have rewritten the analysis methods of PCA, etc., thank you. In line 161-184.
Question 3
In Result some Venn diagrams are shown without being described in Material and Methods how they were constructed, are mentioned of examples of important failures this manuscript presents).
Response: we have modified it in line 206-208.
Question 4
Nevertheless, I think many of these failures are due to the bad quality of English language and the scientific approach for writing the manuscript. I consider not also native English but also a specialist in Science communication ought to revise a new version of the research.
Response:
Thank you for your valuable comment. We have modified it. For example, In the introduction, the distribution and production area of Caoyuanheimo-1 are described, and then the main problems faced by cultivation are put forward, so that the logic is stronger. Then we adjusted the section on methods and results.
Throughout the revision process, it was mainly Professor JI’s work, and she had some experience in edible fungus scientific researches and article writing.
Question 5
Comments on the Quality of English Language
It is too poor.
Response:
Yes, we have done the polish in English at English retouching professional institutions.
Reviewer 2 Report
Comments and Suggestions for Authors
In the study of Caoyuanheimo-1 (Agaricus sp.), the interaction between genes expressed in different stages of fungal development was analyzed through transcriptomic and metabolomic analyses. Differentially expressed genes were identified and the metabolic pathways enriched by these genes in each developmental period were studied, providing insights into gene regulation and key metabolic pathways involved in the growth and development of Caoyuanheimo-1. Furthermore, correlations were observed between differentially accumulated metabolites and gene expression levels, suggesting possible interactions between gene expression and metabolite accumulation at different stages of fungal development.
The article must be accepted for publication, after making the following changes.
Title: Combining transcriptome and metabolome analyzed differentially expressed genes and differential metabolites in development period of Caoyuanheimo-1(Agaricus sp.) from Inner Mongolia of China
Appropriate
Introduction
The introduction requires a further description of the crop of interest regarding its taxonomic classification, its place of origin, the traditional use of this crop, yields and main producing provinces.
Line 55 change fruit to Fruiting body.
The objectives of the research, as well as the working hypothesis, are not clear.
Materials and methods
Section 2.1. Please attach the time periods for the three development periods and remove them from the results.
Mention from which part the carpophore sample was extracted. In addition to mentioning the disinfection technique and management of controlled conditions.
It is mentioned that there were 9 samples, attach this information and eliminate it from the results.
Line 110, (Vanquish, Thermo 110 Fisher Scientific) add state and country of distribution of the equipment.
Why did you not consider using Heterologous Expression or gene expression analysis using RNA-Seq for the study?
Results
The work is very well detailed and very interesting. However, no additional files could be displayed. It should be noted that relevant information is mentioned as a supplementary file necessary to understand the results, which is why as much supplementary information must be attached to the original text.
In section 3.3, accumulated metabolites are mentioned and a reference is given for each of them, please note the base corresponding to this reference.
Figures d and i, the reference values are inverted, please correct them so that they follow the same dynamics as the other images.
Images should improve clarity, please change to a format of 300 dpi or higher.
Consider whether in the results of cumulative metabolites and differential metabolites, it is appropriate to give the difference in terms of percentage in the writing.
Lines 427-429, move on to discussion, since they are in the results section.
Discussion
Line 479-481, the increase in carbon metabolism is mentioned and energy represents vigorous growth and development, however it should be discussed a little more thoroughly. The effects on cellular respiration and CO2 accumulation can influence the acid-base balance in cells and affect cellular respiration. An increase in CO2 levels can stimulate aerobic respiration to remove excess CO2, which could lead to an increase in the production of ATP and respiratory metabolites. Please attach and further discuss this important factor.In my opinion, crop variables were missing, such as the CO2 concentration at each stage, the light/dark hour ratio and pH, since, with CO2 and humidity, this dissociates into water to form carbonic acid, which can decrease the pH of the medium. This can affect enzyme activity and protein function, as many enzymes have optimal pH ranges for their activity. A small discussion section regarding this topic should also be included along with your findings.
Line 491-492 mentions that the metabolites D-xylitol, guanosine and 492 3-hydroxybutyric acid have the same genes associated with them, however, the presence of the same genes associated with different metabolites may be the result of interconnected metabolic pathways, enzyme promiscuity, coordinated gene regulation, or a chance coincidence in the available data. Please thoroughly analyze and discuss this genetic regulation further with your found data.
Conclusions
It is not presented in the document. It is recommended to put the most relevant part of the research.
Literature cited
The references present different forms of citation, please correct.
Examples:
1. Moussa, A.Y.; Xu, B. A narrative review on inhibitory effects of edible mushrooms against malaria and tuberculosis-the 514 world's deadliest diseases. Food Sci. Hum. Wellness. 2023, 12, 17.
2. Zhao, J.Y.; Ding, J.H.; Li, Z.H.; Dong, Z.J.; Feng, T.; Zhang, H.B.; Liu, J.K. Two new sesquiterpenes from cultures of the basid- 516 iomycete Agaricus arvensis. J Asian Nat Prod Res. 2013, 15, 305-309.
3. Zhang, C.B.; Sun, H.X, Gong, Z.J, Zhu, Z.R, Plant terpenoid natural metabolism pathways and their synthases. Plant Physiology Communications. 2007, 43, 779-786.
4. Ma, J, Ding, P.; Yang, G.X.; He, G.Y. Advances on the plant terpenoid isoprenoid biosynthetic pathway and its key enzymes. Biotechnology Bulletin. 2006, 22-30.
Author Response
Thank you very much for your comments and suggestions. Please see the attachment for the answers. Thank you again.

Reviewer 3 Report
Comments and Suggestions for Authors
Line by line suggestions are made below to help with the overall meaning
Line 27-30 -delete or change - this has no real meaning or value
Line 42-44 - delete or reword - this has not real meaning or value
Line 160-172 - move this entire paragraph to the Methods section. Also need to explain the media more precisely.
Figure 2 legend - Need to define HY, HC, and HZ in the legend. Move the Venn diagram description to 2C and the differential DEG description to 2D.
Figure 2b - text is too small, even on the original
Figure 3a, d, g - text is too small, even on the original
Figure 5A - text should be bold or larger
Line 441 - delete - 'The materials used in this article have not been published before and are'. Change to Caoyuanheimo is a unique edible..'
Conclusions - need to be expanded with comparisons to other mushrooms/Basidiomycetes.
Line 510 - need concluding statement
Comments on the Quality of English Language
The English is a problem throughout the paper. This needs to be carefully edited by a native English speaker. A few examples are listed below.
Line 11 - change 'belonged' to 'belonging'
Line 12 - change 'with the value of' to 'this is both'
Line 13-14 change to 'To elucidate the growth and...'
Line 20 change to '..amino acids. '
delete 'and there are certain relationships between differential metabolites and different genes, like'
Line 21 - change to 'Succinic acid is significantly positively correlated with the...'
Line 45 - change 'The study has' to 'Studies have'
Line 48 - delete 'It has been confirmed that'
Line 49 - delete 'it is reported that'
Line 68-69 - please fix grammar - the sentence is not comprehensible
Line 102 - delete 't'
Line 150-152 - This is a sentence fragment
Line 444 - change to '..[48]; its taxonomic..'
Author Response

(The authors gave the same response as above.)

Round 2
Reviewer 1 Report
Comments and Suggestions for Authors
All suggestions have been taken into account by authors. Therefore, there are some typing mistakes (for instance, space between paranthesis and words when describing statistical analysis in Material and Methods) that must be corrected.
Comments on the Quality of English Language
OK
Author Response
Question:All suggestions have been taken into account by authors. Therefore, there are some typing mistakes (for instance, space between paranthesis and words when describing statistical analysis in Material and Methods) that must be corrected.
Response: Thank you for your valuable comment. We have reviewed and revised the relevant parts.
Reviewer 3 Report
Comments and Suggestions for Authors
Most of the comments were addressed. Two comments were not fixed and must be addressed.
Figure 2 legend needs to define HY, HZ, and HC.
Figure 3 legend needs to define HY, HZ, and HC. Figure 3 legend must change so that 3C is the Venn diagram and 3D is the differential DEG diagram.
Author Response
Question1:Figure 2 legend needs to define HY, HZ, and HC.
Response: Sorry and thank you. We have added the relevant definitions.
Question2:Figure 3 legend needs to define HY, HZ, and HC. Figure 3 legend must change so that 3C is the Venn diagram and 3D is the differential DEG diagram.
Response: Yes, you are right. We have changed them and added the definitions of HY, HZ, and HC.